# The health impacts of the COVID-19 pandemic on adults who experience imprisonment globally: A mixed methods systematic review

Hannah Kim[1,2], Emily Hughes[3], Alice Cavanagh[1,3], Emily Norris[3], Angela Gao[3], Susan J. Bondy[4], Katherine E. McLeod[5], Tharsan Kanagalingam[5,6], Fiona G. Kouyoumdjian[5]*

1 Faculty of Health Sciences, McMaster University, Hamilton, Ontario, Canada, 2 Faculty of Medicine, The University of British Columbia, Vancouver, British Columbia, Canada, 3 Michael G. DeGroote School of Medicine, McMaster University, Hamilton, Ontario, Canada, 4 Dalla Lana School of Public Health, University of Toronto, Toronto, Ontario, Canada, 5 Department of Family Medicine, McMaster University, Hamilton, Ontario, Canada, 6 Schulich School of Medicine and Dentistry, University of Western Ontario, London, Ontario, Canada

* kouyouf@mcmaster.ca

**Data Availability Statement:** All relevant data are within the paper and its Supporting information files.

## Abstract

### Background

The prison setting and health status of people who experience imprisonment increase the risks of COVID-19 infection and sequelae, and other health impacts of the COVID-19 pandemic.

### Objectives

To conduct a mixed methods systematic review on the impacts of the COVID-19 pandemic on the health of people who experience imprisonment.

### Data sources

We searched Medline, PsycINFO, Embase, the Cochrane Library, Social Sciences Abstracts, CINAHL, Applied Social Sciences Index and Abstracts, Sociological Abstracts, Sociology Database, Coronavirus Research Database, ERIC, Proquest Dissertations and Theses, Web of Science, and Scopus in October 2021. We reviewed reference lists for included studies.

### Study eligibility criteria

Original research conducted in or after December 2019 on health impacts of the COVID-19 pandemic on adults in prisons or within three months of release.

### Study appraisal and synthesis methods

We used the Joanna Briggs Institute's Critical Appraisal Checklist for Qualitative Research for qualitative studies and the Joanna Briggs Institute's Critical Appraisal Checklist for

**Funding:** The authors received no specific funding for this work.

**Competing interests:** The authors have declared that no competing interests exist.

Studies Reporting Prevalence Data for quantitative studies. We qualitized quantitative data and extracted qualitative data, coded data, and collated similar data into categories.

## Results

We identified 62 studies. People in prisons had disproportionately high rates of COVID-19 infection and COVID-19 mortality. During the pandemic, all-cause mortality worsened, access to health care and other services worsened, and there were major impacts on mental wellbeing and on relationships with family and staff. There was limited evidence regarding key primary and secondary prevention strategies.

## Limitations

Our search was limited to databases. As the COVID-19 pandemic is ongoing, more evidence will emerge.

## Conclusions

Prisons and people who experience imprisonment should be prioritized for COVID-19 response and recovery efforts, and an explicit focus on prisons is needed for ongoing public health work including emergency preparedness.

## Prospero registration number

239324.

## Introduction

From early in the COVID-19 pandemic, community advocates and public health experts have been sounding the alarm about the urgent public health risks of COVID-19 in prisons [1–5]. While attention has primarily focused on preventing the introduction and transmission of COVID-19 in prisons, the intersecting social and structural dynamics of transmission and public health responses suggest a much broader scope of impact.

The risk of COVID-19 introduction into prisons is high because of frequent movement of people between prisons and the community, including people being admitted to prison, people on intermittent sentences, and staff. Prison environmental conditions, such as close quarters, overcrowding, and limited individual autonomy over prevention measures increase the risks of transmission [2, 4, 6, 7]. The high prevalence of chronic health conditions among people in prison, including respiratory diseases, cardiovascular diseases, and conditions associated with immune compromise [8–11], increases the risk of serious sequelae of COVID-19 infection in this population.

The COVID-19 pandemic also affects population health status through mechanisms other than COVID-19 infection. While many countries reduced their prison population size through measures to reduce entry into custody and increase release from custody [12, 13], there may not have been commensurate increases in discharge planning or community resources to support needs, including for treatment beds and shelter beds. In addition, the transition to remote services and reduced scope and hours of services in many jurisdictions may have limited access to essential health and social services [14]. Anxiety about the pandemic, isolation, difficulties

navigating public health measures, fewer employment opportunities, and reduced health and social supports may all contribute to worse mental health and increased substance use among people who experience imprisonment [14, 15]. In prisons in particular, efforts to mitigate transmission have included increased time in cells, restrictions in work and education programs, and limited visits and social interactions [13], all of which negatively impact mental health [16]. There is also evidence from some jurisdictions that the pandemic has been associated with an increasingly unstable and toxic illicit drug supply [15, 17], which may further increase the already substantially elevated risk of overdose for people who experience imprisonment [18], both while in prison and in the community post-release.

The health impacts of the COVID-19 pandemic have exacerbated existing health inequities shaped by broader patterns of marginalization and colonization for people who experience imprisonment [19]. For example, people with clinical conditions such as substance use disorders and mental illness are overrepresented in prisons [8]. In addition, certain demographic groups are overrepresented in prisons, for example, in Canada, people who are Indigenous and Black [20, 21]. This overrepresentation means that the harms of the COVID-19 pandemic experienced by people who experience imprisonment disproportionately impact specific communities.

A comprehensive review of current knowledge of the impacts of the pandemic on people who experience imprisonment is essential to inform ongoing COVID-19 prevention and response, pandemic recovery, and emergency preparedness, and to address persistent health and healthcare inequities. To address this gap, we conducted a mixed methods systematic review of evidence about the impact of the COVID-19 pandemic on the health of people who experience imprisonment.

## Methods

### Protocol and registration

We developed a research protocol, which we registered in PROSPERO under registration number CRD42021239324.

### Search

We developed a search strategy in consultation with a research librarian. We searched Medline, PsycINFO, Embase, the Cochrane Library, Social Sciences Abstracts, CINAHL, Applied Social Sciences Index and Abstracts, Sociological Abstracts, Sociology Database, Coronavirus Research Database, ERIC, Proquest Dissertations and Theses, Web of Science, and Scopus, and the search strategy is available in S1 File or at https://www.crd.york.ac.uk/PROSPEROFILES/239324_STRATEGY_20210224.pdf. We reviewed reference lists of included studies and relevant reviews. We limited our search to studies appearing since December 2019, when the first human cases of COVID-19 were identified. We did not have any language restrictions for eligible articles, though we used only English language search terms.

We initially ran the database search on February 28th, 2021. We updated the search on October 14th, 2021.

For any identified articles that were pre-prints, we searched for a published peer-reviewed version of the article, and if a published peer-reviewed document was found (through searches in Google or Medline, or after contacting the corresponding author), we updated the extracted data and the reference based on the published peer-reviewed article. If a published peer-reviewed document was not available, we used the published pre-print.

## Eligibility criteria

Studies were eligible for inclusion if they met four criteria. The first criterion was a focus on adults aged 18 and older in prisons, jails, or correctional facilities or within 3 months of release from imprisonment; we use the term "prison" in this article to describe all these carceral settings, though for each study we describe the specific setting. If studies included data for youth in detention or other populations in addition to adults in prisons, we included the study only if data were stratified for adults experiencing imprisonment. The second criterion was research that occurred in or after December 2019, in order to exclude studies of other coronaviruses. The third criterion was original quantitative or qualitative research, which could include surveillance and outbreak data. The fourth criterion was health impacts of the COVID-19 pandemic. We defined health impacts as health outcomes directly and/or indirectly attributable to the COVID-19 pandemic, including outbreaks, changes in health care and other services related to or coincident with the pandemic, COVID-19 morbidity and mortality, and changes in mental or physical health coincident with the pandemic.

We excluded studies if they focussed only on immigration detention facilities, or if they described only prison policies, which may not align with practice or experience.

## Selection criteria and data extraction

Two authors independently reviewed all titles and abstracts identified in the search. We resolved discrepancies through discussion regarding whether the article was eligible for full text review based on the title and abstract. Two authors independently reviewed each full article for eligibility for inclusion. Disagreements in the decisions by the two reviewers were resolved through discussion, and the involvement of a third reviewer when necessary.

We used a data extraction form, which we modified based on the Joanna Briggs Institute (JBI) extraction tool in JBI SUMARI. We extracted study information, study characteristics, participants, methods, and outcomes. When studies presented data for correctional staff as well as people in custody, we extracted only data for people in custody. We transformed quantitative data by "qualitizing" extracted quantitative data, i.e. converting quantitative data into textual descriptions of the findings [22]. For qualitative data, we extracted themes or subthemes with corresponding illustrations.

## Synthesis

We planned *a priori* to use a convergent integrated approach to combine extracted data from quantitative studies and qualitative studies, which involves assembling the qualitized and qualitative data, and categorizing and pooling together these data based on similarity in meaning to produce a set of integrated findings [22]. Three authors independently coded the data, and met iteratively to review and discuss emerging codes (or groupings), and then two authors re-coded the data with the established coding framework. For most codes, we found qualitized data or qualitative data but not both types of data, so we could not pool across types of data. For some categories there were limited data, so we present the findings as a narrative synthesis [22]. When relevant, we reported data on the type of carceral setting, period of the study, and region of the study, assuming these factors may be relevant to the interpretation of the data presented, and also since epidemiological factors (such as COVID-19 variant) and social and political factors varied over time and across regions.

## Risk of bias

We appraised the risk of bias in each study using the Joanna Briggs Institute's Critical Appraisal Checklist for Qualitative Research for qualitative studies and the Joanna Briggs

Institute's Critical Appraisal Checklist for Studies Reporting Prevalence Data for quantitative studies [22]. We did not exclude studies on the basis of quality.

## Results

### Search results

As shown in Fig 1, we identified 2,092 unique references, 1,995 of which were assessed as irrelevant on title and abstract screen. We reviewed 97 full text studies for eligibility: 34 were irrelevant and 1 study was excluded because it had been retracted. Ultimately, 62 studies were eligible for inclusion in this review.

### Characteristics of included studies

Studies were mostly conducted in the USA (n = 40), while 9 were conducted in Europe, 7 in the United Kingdom, 4 in Brazil, and 2 in Canada (Table 1). We included 54 quantitative studies and 8 qualitative studies; only 1 qualitative study was conducted in the USA. Almost half the studies (n = 28) were conducted in the first 6 months of 2020.

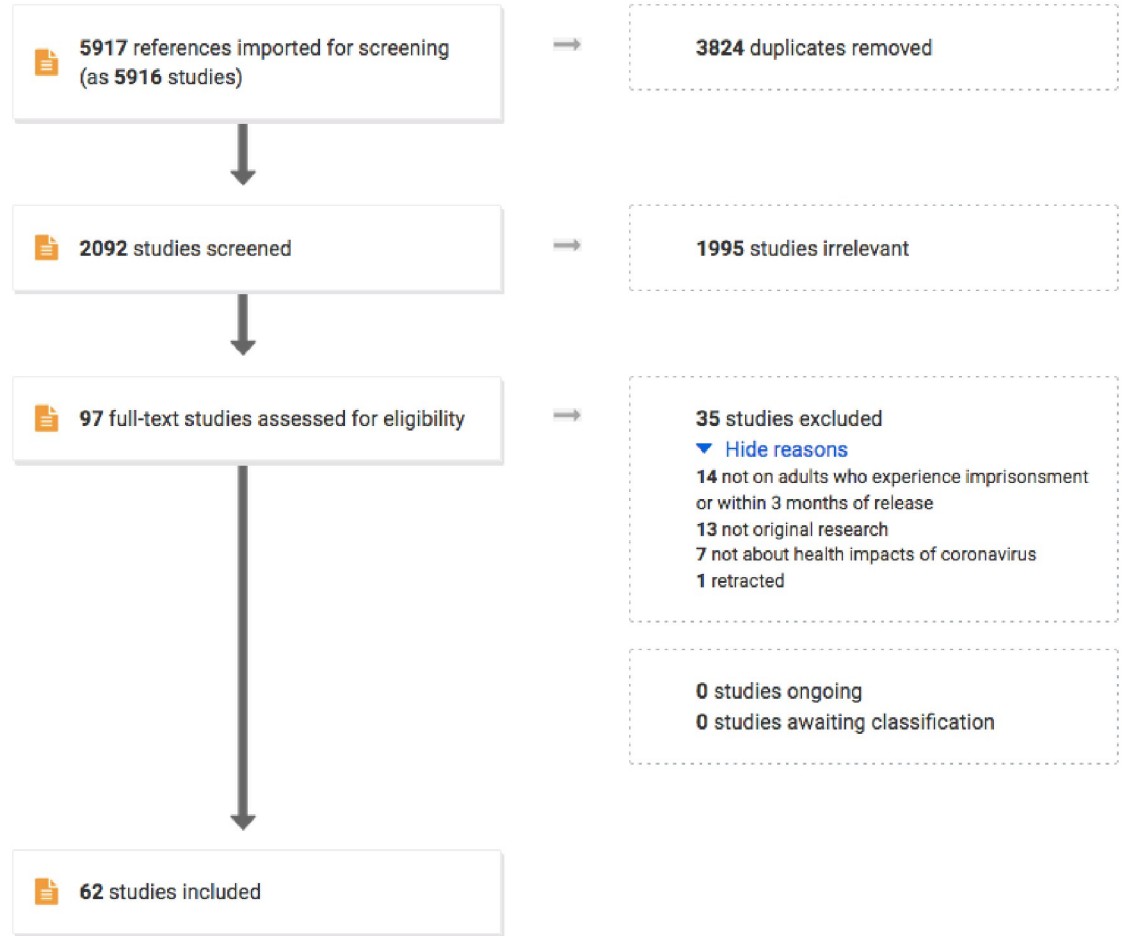

**Fig 1. Flow diagram for studies included in a systematic review on the health impacts of the COVID-19 pandemic on people who experience imprisonment.**

**Table 1. Characteristics of studies included in a systematic review on the health impacts of the COVID-19 pandemic on people who experience imprisonment, N = 62.**

| Country | First author | Region[a] | Period | Type of prison setting |
|---|---|---|---|---|
| Brazil | Crispim 2021 | N/A | April-August 2020 | prisons |
| | Gouvea-Reis 2021 [23] | Brasilia | April-October 2020 | penitentiary |
| | Gouvea-Reis 2021 [24] | Brasilia | April-June 2020 | penitentiary |
| | Ribeiro 2020 | N/A | December 2019-September 2020 | prisons |
| Canada | Blair 2021 | N/A | March-May 2020 | federal prisons |
| | McLeod 2021 | British Columbia | May 2020 | provincial prison |
| Italy | Cerrato 2021 | Bologna | March-June 2020 | prison |
| | Giuliani 2021 [25] | Lombardy | March-July 2020 | prisons |
| | Giuliani 2021 [26] | Milan | February-April 2020 | prison |
| | Sorge 2021 | San Vittore | March-May 2020 | prison |
| | Stufano 2021 | Apulia | November 2020-January 2021 | correctional facility |
| Spain | Marco 2021 [27] | Barcelona | March-April 2020 | prison |
| | Marco 2021 [28] | Barcelona | Not specified | prison |
| Switzerland | Getaz 2021 | Geneva | 2016–2020 | prison |
| United Kingdom | Coleman 2020 | Not specified | 2020 | prison |
| | Gray 2021 | Northern Ireland | April-December 2020 | prisons |
| | Maycock 2021 [29] | Scotland | Not specified | prison |
| | Maycock 2021 [30] | Scotland | Not specified | prison |
| | Maycock 2021 [31] | Scotland | Not specified | prison |
| | Suhomlinova 2021 | England, Wales | April-October 2020 | prisons |
| | Wilburn 2021 | Not specified | March-June 2020 | prison |
| USA | Altibi 2021 | Michigan | March-June 2020 | state prisons |
| | Bandara 2020 | N/A | May 2020 | county jails and state prisons |
| | Berk 2021 | Rhode Island | December 2020-February 2021 | combined jail and prison |
| | Brinkley-Rubinstein 2021 | Rhode Island | March-May 2021 | combined jail and prison |
| | Chan 2021 | New York City, New York | March-April 2020 | jails |
| | Chin 2021 [32] | California | March-October 2020 | state prisons |
| | Chin 2021 [33] | California | December 2020-March 2021 | state prisons |
| | Chin 2022 | California | December 2020-March 2021 | state prisons |
| | Collica-Cox 2020 | New York | March-May 2020 | jail |
| | Dunne 2021 | Idaho | July-November 2020 | state correctional facilities |
| | Hagan 2020 | N/A | April-May 2020 | federal prisons, state prisons, county jails |
| | Hagan 2021 [34] | N/A | December 2020-April 2021 | federal prisons |
| | Hagan 2021 [35] | Texas | July-August 2021 | federal prison |
| | Hershow 2021 | Wisconsin | August-October 2020 | state prison |
| | Jimenez 2020 | Massachusetts | April-July 2020 | jails and state prisons |
| | Kennedy 2020 | Connecticut | March-June 2020 | prisons and jails |
| | Khairat 2021 | North Carolina | June-November 2020 | prisons |
| | KhudaBukhsh 2021 | Ohio | April 2020 | prison |
| | Lehnertz 2021 | Minnesota | March-June 2020 | correctional facilities |

*(Continued)*

**Table 1.** (Continued)

| Country | First author | Region[a] | Period | Type of prison setting |
|---|---|---|---|---|
| | Leibowitz 2021 | Massachusetts | April 2020-January 2021 | state prison |
| | Lemasters 2020 | USA and Puerto Rico | March-July 2020 | prisons |
| | Lewis 2021 | Utah | September 2020-January 2021 | correctional facility |
| | Maner 2021 | N/A | April 2020-January 2021 | prisons |
| | Marquez 2021 [36] | Texas | April 2019-March 2021 | carceral settings |
| | Marquez 2021 [37] | Florida | 2015–2020 | prisons |
| | Njuguna 2020 | Louisiana | May 2020 | correctional and detention facility |
| | Nowotny 2020 | USA and Puerto Rico | April-July 2020 | prisons |
| | Pettus-Davis 2021 | Midwest and Southeast states | March-November 2020 | state correctional facilities |
| | Pocock 2020 | Arizona | April 2020 | correctional facility |
| | Puglisi 2021 | Not specified | prior to May 2020 | jail |
| | Pyrooz 2020 | Oregon | April-May 2020 | prison |
| | Saloner 2020 | N/A | March-June 2020 | state and federal prisons |
| | Stern 2021 | Washington, California, Florida, Texas | September-December 2020 | prisons and jails |
| | Toblin 2021 | N/A | February-September 2020 | federal prisons |
| | Tompkins 2021 | Arkansas | April-May 2020 | correctional facility |
| | Vest 2021 | Texas | March-July 2020 | prisons |
| | Wadhwa 2021 | Chicago | May 2020 | correctional facility |
| | Wallace 2020 | N/A | January-April 2020 | state prisons, federal prisons, detention facilities |
| | Wallace 2021 | Louisiana | May-June 2020 | detention centre |
| Zawitz 2021 | Illinois | March-April 2020 | jail | |
| Multiple countries | Montanari 2021 | Europe | March-June 2020 | prisons |

[a]If applicable and specified.

## Risk of bias

For the 54 quantitative studies (S1 Table), 29 did not satisfy the quality indicator in at least one domain, suggesting risk of bias, and 4 did not provide sufficient information on all relevant domains to be able to appraise quality for all domains of interest. For the 8 qualitative studies (S2 Table), all studies did not satisfy the quality indicator in at least one domain and 4 studies did not contain sufficient information to be able to appraise quality on all domains of interest.

## Data synthesis

We grouped data into three categories with several subcategories, as shown in Table 2.

## Burden of COVID-19 infection

**COVID-19 cases.** *Population incidence rates.* Five studies reported population-level incidence data for people incarcerated in US prisons. Data collected from March to July 2020 showed that 34 of the 53 prison systems had higher incidence rates than those for the general population, and for six states, the cumulative incidence rate was more than 100 cases per 1,000 higher for the prison population [38]. By June 6[th], 2020, the cumulative incidence rate for people in federal and state prisons was 32.5 per 1,000, which was 5.5 times higher than the whole

**Table 2. Coding framework for systematic review on the health impacts of the COVID-19 pandemic on people who experience imprisonment.**

| Burden of COVID-19 infection | COVID-19 cases | Population incidence rates |
| --- | --- | --- |
| | | Percent positivity |
| | | Number of cases |
| | | Factors associated with infection |
| | | Symptom status of cases |
| | COVID-19 hospitalizations | Proportion of cases hospitalized |
| | | Factors associated with hospitalization |
| | COVID-19 death and mortality | Population mortality rates |
| | | Case fatality |
| | | Factors associated with death |
| COVID-19 prevention strategies | Primary prevention | Vaccination |
| | | Hygiene |
| | | Quarantine and isolation |
| | Secondary prevention | Testing |
| Other impacts of COVID-19 pandemic on health status | All cause-mortality | |
| | Changes to services | |
| | Impacts on relationships with family and staff | |
| | Impacts on mental health | |

population rate of 5.9 per 1,000 [39]. By September 23[rd], 2020, the cumulative incidence rate for people in federal prisons was 11,710.1 per 100,000, compared with 2,484.4 cases per 100,000 for adults in the general population [40]. In Massachusetts, the cumulative incidence rate in July 2020 was 2.9 times higher for people in jails and state prisons (44.3/1,000) compared with the general population [41], and in people in state prisons the incidence rate by January 11[th], 2021 was 956 cases per 100,000 person-weeks, compared with 150 cases per 100,000 person-weeks for the general population [42].

Studies in Canada, Brazil, and Italy similarly showed high incidence rates for people in prisons. The cumulative incidence for the federal prison population in Canada was 10 times higher in Quebec, 2 times higher in Ontario, 6 times higher in British Columbia compared with the general population [43]. In Brazil, a survey of state penitentiaries identified 23,054 confirmed cases of COVID-19 by September, 2020, for a cumulative incidence rate of 30.9 per 1,000, compared with 19.7 per 1,000 for the total population [44].

*Percent positivity*. Three studies reported on cases identified and percent positivity based on enhanced testing initiatives. In 16 US facilities that conducted mass testing between April and May 2020, the incidence ranged from 0% to 86.8%, with a median of 29.3%, and mass testing increased total known cases from 642 before mass testing to 8,239 after testing [45]. In an Italian correctional facility for people with chronic diseases, between November 2020 and January 2021, 2 people tested positive of 426 people tested in an initial mass testing campaign (0.5%), and 0 people tested positive of 480 in a second mass testing campaign from December 2020 to January 2021 (0%) [46]. In state correctional facilities with work release programs in Idaho, USA, the percent positivity in mass testing between July and November 2020 ranged from 1–92% [47].

Sixteen studies reported the percentage of people who tested positive in the context of outbreaks [23, 26–28, 35, 48–58], as shown in Fig 2, with a wide range within and across countries from 2.5% to 90.6%, but with very high proportions of people testing positive in most outbreaks.

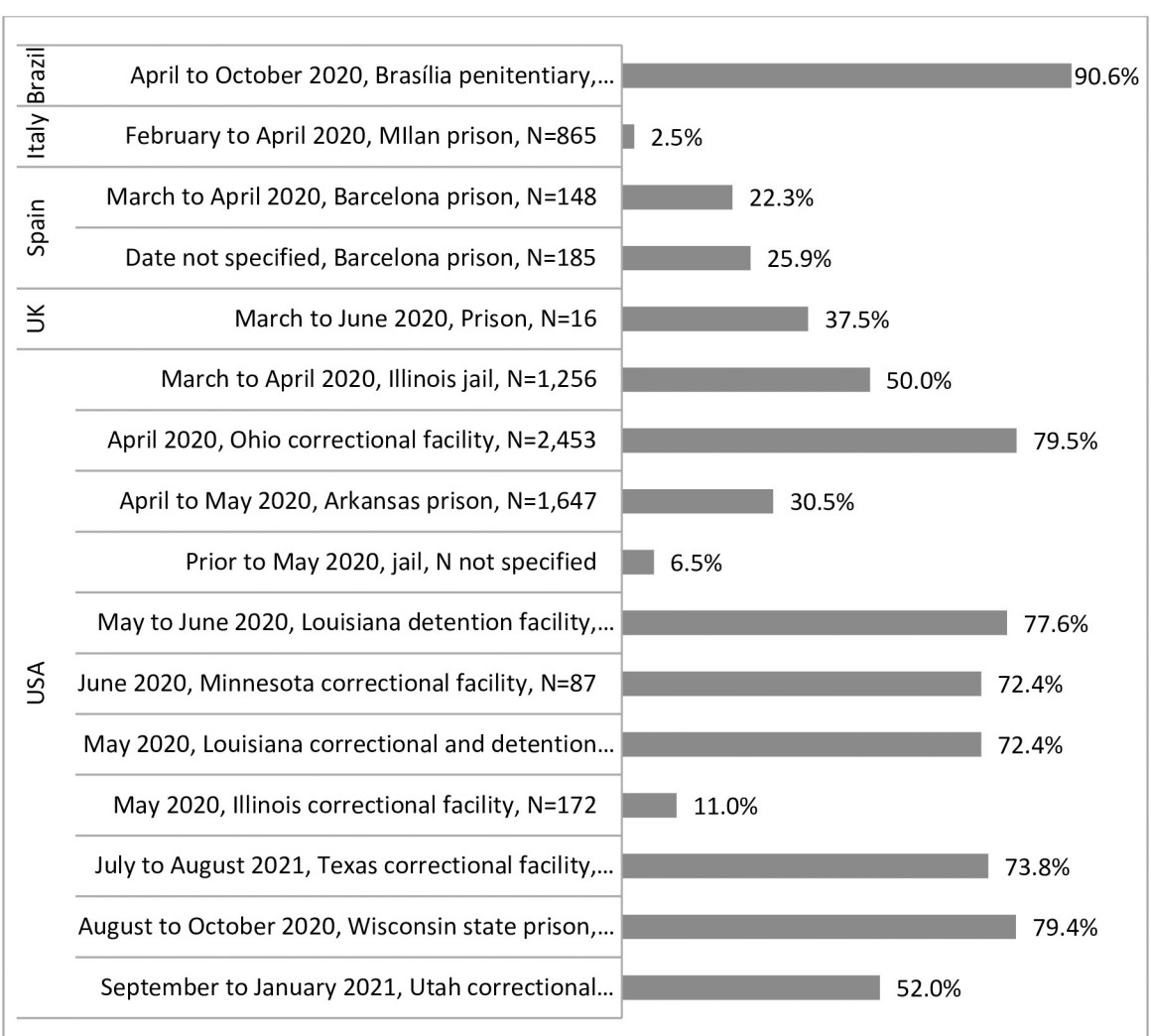

**Fig 2. Percent testing positive in outbreaks in studies included in a systematic review on the health impacts of the COVID-19 pandemic on people who experience imprisonment, by country.**

Eight studies presented test positivity data outside of enhanced testing initiatives or outbreaks. In New York City, USA jails, 58% people tested positive of 978 people tested from March to April 2020 [59]. In prisons and jails in Connecticut between March and June 2020, 12% (n = 1,240) of people tested positive of the 10,304 people tested for COVID-19 [60]. A study of 53 US prison systems from March to July 2020 found that percent positivity varied widely from 0% to 42%, and in most states, percent positivity was higher for those in prison compared with the general population [38]. In 18 prisons in Lombardy, Italy, 102 cases were identified in 5,777 people tested during the first COVID-19 wave between March and July 2020, for an incidence of 2.7% [25]. A Massachusetts, USA study found 5% test positivity for 8,455 tests conducted in people in state prisons and 14% test positivity in 1,843 tests conducted in county jails [41]. In California, USA, state prisons, test positivity was 19.2% by October 2020 [32]. Between March and November 2020, 9.9% of 317 people across 38 correctional facilities in the Midwest and Southeast USA had had confirmed COVID-19 [61]. In a Rhode

Island, USA, correctional facility from March to May 2021, the COVID-19 incidence was 1.3% (20/1,539) for people who were vaccinated [62].

*Number of cases*. Eight studies reported on the number of cases without providing denominators. A US study of surveillance data from 32 state and territorial health department jurisdictions found that there were 4,893 COVID-19 cases among incarcerated or detained people between January 21[st] and April 21[st], 2020 [63]. In a prison outbreak in northern Italy between March and June 2020, 34 cases were identified [64]. Between March and July 2020, there were 11,799 confirmed COVID-19 cases among incarcerated people in the Texas Department of Criminal Justice [65]. In a penitentiary outbreak in Brasília, Brazil from April to June 2020, 859 cases were identified [24]. In Brazil, between April and August 2020, a total of 18,767 COVID-19 cases were identified among incarcerated people [66]. A European study reported 26 confirmed COVID-19 cases in Belgium between March and June 2020, 119 cases in France by mid-May 2020, and 315 cases in Italy between March 9 and May 2020 [67]. In state correctional facilities with work release programs in Idaho, USA, there were 382 cases identified between July and November 2020 [47]. In a UK prison outbreak in 2020, there were 88 possible, probable, or confirmed cases [68].

*Factors associated with infection*. Regarding factors associated with COVID-19 infection, a study in a correctional facility in Arkansas, USA, in April and May 2020 found that pre-existing chronic lung disease was associated with infection, but other pre-existing medical conditions (hypertension, diabetes, cardiovascular disease, chronic kidney disease, chronic liver disease) were not [49]. In a prison in Barcelona, Spain, age, history of diabetes, and HIV infection, respectively, were not associated with infection [27]. In an outbreak in a Texas prison in July and August 2021, vaccination status and diabetes were associated with infection [35].

*Symptom status of cases*. Several studies described the proportion of cases that were symptomatic and the proportion of cases with specific symptoms. The proportion of cases identified as symptomatic was 89.8% (510/568) in New York City, USA jails from March to April 2020 [59], 76.3% (479/628) in an outbreak in an Illinois, USA jail in March and April 2020 [50], 4% (3/71) in a Louisiana, USA correctional facility in outbreak in April to May 2020 [52], 18.8% in a men's correctional facility in Arkansas, USA in April and May 2020 [49], 43% (48/111) in a Louisiana, USA detention facility outbreak in May and June 2020 [53], 37% (12/19) in a correctional facility in Illinois, USA in May 2020 [56], and 53.5% (68/127) for people aged 60 and older in an outbreak in a penitentiary in Brazil from April to October 2020 [23].

In a men's correctional facility in Arkansas, the most common symptoms identified for cases were headache, runny nose, chills, and cough, however, each of these symptoms was reported by less than or equal to 6% of cases [49]. In the correctional facility in Illinois, USA, the most commonly reported symptoms for symptomatic cases were loss of taste or smell (47%), headache (32%), and chills (26%) [56]. In the Louisiana jail, most commonly reported symptoms for cases were headache (32%), loss of taste or smell (31%), and nasal congestion (26%) [53]. In the outbreak penitentiary in Brazil, the most common symptoms in cases were headache (34.9%), followed by cough (30.2%), fever (28.9%), ageusia/anosmia (19.7%), dyspnea (16.7%), myalgia (10.5%), sore throat (8.0%), nasal congestion (5.7%), and diarrhea (4.0%) [24].

**COVID-19 hospitalizations.** *Proportion of cases hospitalized*. Several studies presented data on the percentage of cases hospitalized, ranging from 0.0% to 10.0% [24, 27, 32, 35, 47, 53–55, 57, 59, 60, 63, 68], as shown in Fig 3.

Some studies also provided data on the percentage of cases admitted to ICU: 0.3% of 13,636 cases in state prisons in California, USA, between March and October 2020 [32], 1.6% (20/1,240) in Connecticut, USA state prisons from March to June 2020 [60], 1.4% (8/568) in New

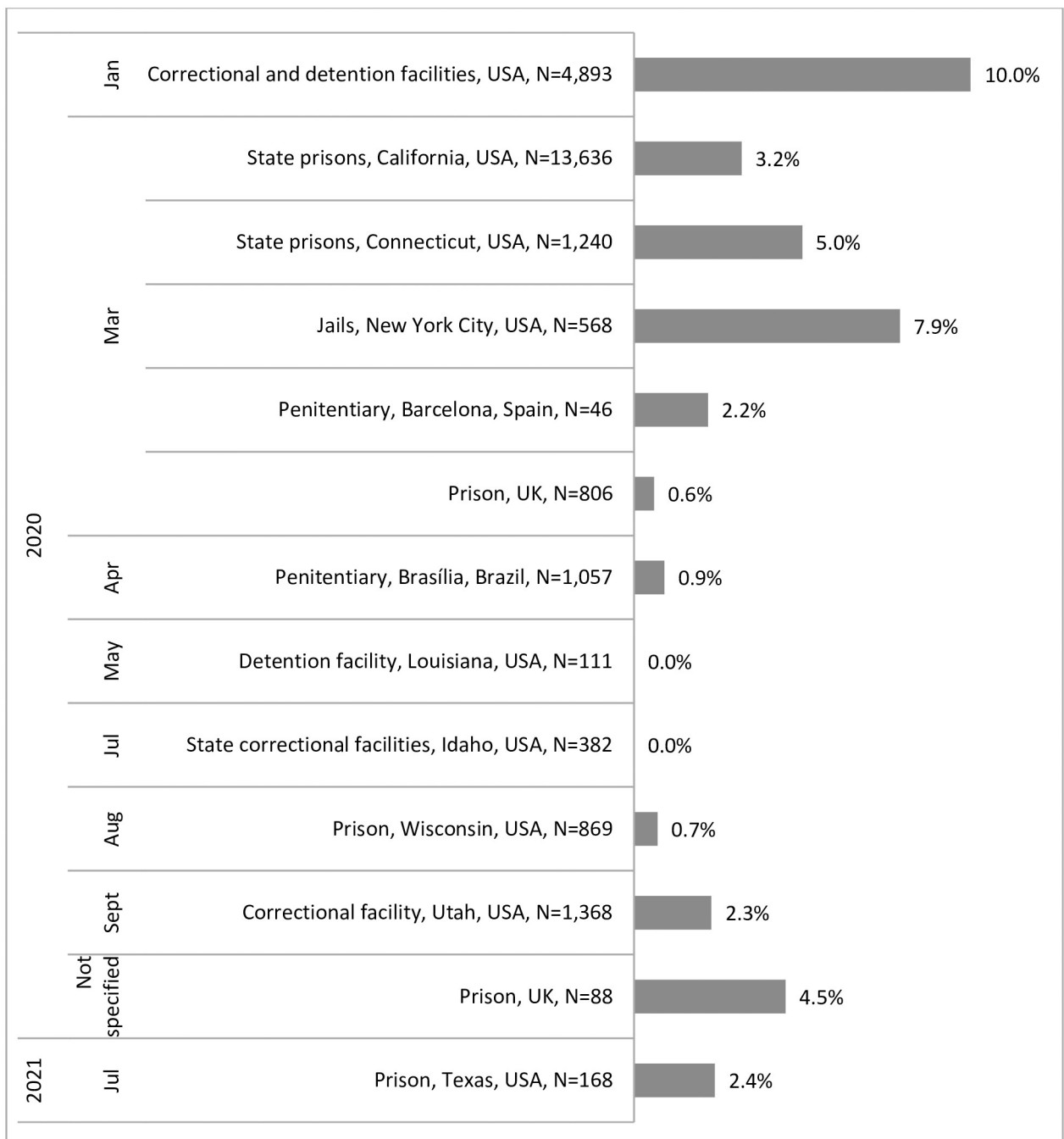

**Fig 3. Percent of COVID-19 cases hospitalized for people who experience imprisonment in studies included in a systematic review on the health impacts of the COVID-19 pandemic on people who experience imprisonment, by month of start of study period.**

York City, USA jails in March and April 2020 [59], and 0.0% (0/46) in a penitentiary in Barcelona, Spain [27].

A Michigan, USA study of people hospitalized with COVID-19 found that a higher proportion of those who were incarcerated were admitted to ICU (29.6%) compared with the proportion of people who were not incarcerated who were admitted to ICU (18.7%), and the median

**Table 3. Estimates of COVID-19 mortality for people in prison relative to the general population, controlling for age and sex, in studies included in a systematic review on the health impacts of the COVID-19 pandemic on people who experience imprisonment[a].**

| Indicator | Location | Period and demographic group | | Point estimate |
|---|---|---|---|---|
| COVID-19 standardized mortality ratio | US prisons [70] | April 25, 2020 week | | 1.2 |
| | | May 2, 2020 week | | 1.6 |
| | | May 9, 2020 week | | 2.0 |
| | | May 16, 2020 week | | 2.1 |
| | | May 23, 2020 week | | 2.2 |
| | | May 30, 2020 week | | 2.3 |
| | | June 6, 2020 week | | 2.4 |
| | | June 13, 2020 week | | 2.4 |
| | | June 20, 2020 week | | 2.5 |
| | | June 27, 2020 week | | 2.6 |
| | | July 4, 2020 week | | 2.7 |
| | | July 11, 2020 week | | 2.8 |
| | US federal prisons [40] | February to September 2020 | All | 2.6 |
| | | | Males | 2.5 |
| | | | Females | 4.6 |
| | Texas carceral settings [36] | April 2020-March 2021 | Black | 25.4 |
| | | | Hispanic | 33.2 |
| | | | White | 16.4 |
| COVID-19 relative mortality rate | US federal and state prisons [39] | March-June 2020 | | 3.0 |

[a]Data for the same prisons and time period may have been included in multiple studies.

time from self-reported onset of symptoms to hospital admission was longer for people who were incarcerated (6 days) compared with those who were not incarcerated (5 days) [69].

*Factors associated with hospitalization*. In two studies, older age was associated with hospitalization and ICU admission in cases [32, 60]. In prisons and jails in Connecticut, heart disease was the strongest predictor of hospitalization for those who tested positive for COVID-19, and heart disease, age, and autoimmune disease were each associated with ICU admission [60]. In an outbreak in Texas, USA from July to August 2021, the proportion of cases hospitalized was 8% for those unvaccinated (3/39) and 1% of those fully vaccinated (1/129) [35].

**COVID-19 deaths and mortality.** *Population mortality rates*. Surveillance and health administrative data show high COVID-19 mortality rates, i.e. deaths from COVID-19 per population (not per case), and large numbers of COVID-19 deaths in prisons. By July 2020, 32 out of 50 US state departments of corrections had reported at least one COVID-19 related death [70], and for those states, 10 reported little to no difference in mortality between the prison and general populations, 6 reported a substantially higher mortality rate in the general population, and 3 reported a slightly higher mortality rate in the prison population [70]. In Texas prisons, 120 people died from COVID-19 by July 15[th], 2020, for a COVID-19 crude mortality rate (CMR) of 79.4 per 100,000, compared with 11.8 per 100,000 for the general population [70]. Data from Brazil between December 2019 and September 2020 showed the COVID-19 CMR was 14.5 per 100,000 for people in prison compared with 60.4 per 100,000 for the whole population [44], with substantial variation in COVID-19 mortality rates for people in prison across states.

Four US studies compared COVID-19 mortality rates for people in prison compared with people in the general population after controlling for age and sex [36, 39, 40, 70], as shown in Table 3.

*Case fatality*. Several US studies reported the proportion of cases that died: 0.5% (3/568) in New York City jails from March to April 2020 [59], 1.1% (7/628) in a jail in Illinois from March to April 2020 [50], 0.6% (7/1,240) in prisons and jails in Connecticut from March to June 2020 [60], ranging from 0% to 0.9% based on surveillance data for prison systems from March to July 2020 [38], 0.5% in state prisons in California from March to October 2020 [32], 0% of 382 cases in Idaho, USA between July and November 2020 [47], and 0.7% in a Utah outbreak from September 2020 to January 2021 [55]. A European study reported 0 deaths of 26 cases (0%) in Belgium between March and June 2020, 1 death of 119 cases (0.8%) in France by mid-May 2020, and 4 deaths in 315 cases (1.3%) in Italy between March 9th and May 4th, 2020 [67]. In Quebec, Ontario and British Columbia, Canada, the case fatality ratio in federal prisons was higher than that of the general population between March and May 2020: 0.5% compared with 0.3% [43].

In Michigan, USA, in-hospital all-cause mortality was higher in incarcerated individuals with COVID-19 (29.6%) compared with people with COVID-19 in the general population (20.1%); after controlling for age, sex, obesity, and comorbidities, the adjusted odds ratios for those incarcerated compared to those not incarcerated was 2.3 for in-hospital mortality and 2.0 for 30-day mortality [69].

*Factors associated with death*. Two additional studies presented data on factors associated with COVID-19 mortality. Data for people in prisons and jails in Connecticut, USA found that older age was a risk factor for death, whereas chronic conditions, other demographic characteristics, and facility type were not [60]. A separate California, USA study identified comorbidities, BMI over 40 and age 65 years or older as mortality risk factors [32].

## COVID-19 prevention strategies

**Primary prevention.** *Vaccination intentions and rates*. One study explored vaccination intentions. In a study of 5,110 people in correctional facilities in Washington, California, Florida and Texas, USA, from September to December, 2020, 44.9% said they would receive a vaccination, 9.7% said they would hesitate, and 45.4% said they would refuse [71]. There was similar willingness to vaccinate reported among men and women, greater willingness for people who were older, and lower willingness for Black people.

Three studies reported vaccination rates. In a correctional facility in Rhode Island, USA, vaccination was offered for 6 weeks between December 2020 and February 2021, and 76.4% of incarcerated individuals received the vaccine; 90.9% of the first group offered the vaccine (age >65 years old, immunocompromised, or age >55 with comorbidities) received the vaccine, and by 4 months later, 77.7% of the incarcerated population was fully vaccinated [72]. There were no significant adverse events reported after vaccination [72]. In California, USA, by March 2021, 49% of people that met inclusion criteria in prisons had received at least one dose of the vaccine, and 22% had received 2 doses [33]. By April 2021, of the total population incarcerated in US federal prisons, 44.8% had received at least one vaccine dose, 29.9% had been fully vaccinated, 69.8% had been offered vaccination, and 64.2% of those offered vaccination had accepted [34]. For 2,514 people who had signed a vaccine declination form and were offered vaccination a second time, 1,415 accepted on the second offer [34].

In three studies in the USA, factors associated with higher vaccination were a history of smoking and Black, non-Hispanic race/ethnicity in Texas [35]; age, female sex, non-Hispanic White compared with non-Hispanic Black or Asian race/ethnicity, being born outside of the USA or with an unknown country of birth, number of medical conditions associated with severe COVID-19 illness, as well as institution type for people in federal prisons [34]; and

older age, medical vulnerability, and Hispanic or White race or ethnic group compared with Black race or ethnic group in people in California state prisons [33].

*Hygiene*. In an Italian prison, people reported an inability to adhere to public health guidance between March and May 2020 [73].

*"Here, in prison, it's not easy to maintain social distances, to have the right masks and to sanitize everything."*

*Quarantine and isolation*. Quarantine data for 9 US states show a wide range in the maximum quarantine rates from April 2020 to January 2021, from 36.3 per 1,000 people (n = 970) in Indiana to 843 per 1,000 people (n = 40,827) in Ohio [74].

A study of people at the time of release from a correctional facility in Arizona, USA in April and May 2020 found that release of cases to a medical recovery site for isolation substantially decreased the number of secondary infections in the community; from 7 released cases leading to 6 secondary cases associated with 4 hospitalizations to 12 released cases leading to 0 secondary infections after implementation [75].

**Secondary prevention.**    *Testing*. Eight studies reported on COVID-19 testing rates. In federal prisons, state prisons, and jails across the USA from April to May 2020, 16,392 incarcerated people were offered testing, representing a range of 2.3% to 99.6% across facilities (median 54.9%) [45]. Testing rates similarly varied widely in a study of 53 US prison systems from March to July 2020, ranging from 6 tests administered per 1,000 incarcerated people in Hawaii to 1,530 tests administered per 1,000 incarcerated residents in Minnesota; in most states, testing rates were higher for people in prison compared with the general population [38]. A study of people in US federal prisons also found higher testing rates in prisons compared with the general population; as of September 2020, 50.3% of people in US federal prisons had been tested, compared to 32.5% of the U.S. population (assuming 1 test per person) [40]. In contrast, from March to April 2020, 64% of federal prisons in Canada (32/50) had lower testing rates than the general population [43]. This was also the case for a jail in Westchester County, New York, USA, in which over 20% of people were tested from March to May 2020, which was lower than New York State's testing rates of 25% in the general population [76]. In New York City, USA jails, 15% (978/6,311) of incarcerated residents were tested from March to April 2020 [59]. A Massachusetts, USA study found that by July 2020, 8,455 tests had been administered in state prisons with 7,735 people in custody, while 1,843 tests were administered in county jails with 7,252 people in custody [41]. In California, USA, state prisons, 81.8% of 96,440 people had been tested for COVID-19 by October 2020 [32]. A separate study in California state prisons identified that testing rates varied by vaccination status and were lower for people who were unvaccinated: in January 2021, the testing rate was 933 tests per 10,000 person-days for unvaccinated people, 1,167 tests per 10,000 person-days for partially vaccinated people, and 2,018 per 10,000 person-days for those fully vaccinated [77].

### Other impacts of the COVID-19 pandemic on health status

**All-cause mortality.**    Three US studies examined changes in mortality during the COVID-19 pandemic. The Departments of Corrections for Delaware, Michigan, New Jersey, and Ohio reported COVID-19 deaths in the first 6.5 months of 2020 in excess of 50% of deaths from all causes for the most recent year for which mortality data were available, which was 2016 [70]. The CMR was significantly higher in the Florida state prison population in 2020 compared with 2019, and life expectancy was significantly lower in 2020 than in any of the other year over the study period (2015–2019), with a life expectancy drop of 4.1 years between

2019 and 2020 for this population [37]. The all-cause mortality rate for incarcerated individuals in Texas increased by 85% from April 1st, 2019 to March 31st, 2021, with a 126% increase in the Black population, a 107% increase in the Hispanic population, and a 52% increase in the White population [36].

**Changes to services.** Seven studies described changes to services and programs. Five studies identified reduced access to health services during the pandemic for people who experience imprisonment, with one participant in a study of incarcerated males in Scotland describing health services as "at a minimum" [29]. For example, a study of transgender women and non-binary individuals in men's prisons in the UK found that during the pandemic, access to health care worsened as staff cared for COVID-19 cases [78].

> *"[N]o dentist, no opticians, no diabetic clinic, no asthma clinic, and only essential doctor appointments[.]"*

In a men's prison in Oregon, USA, 71% of incarcerated individuals reported that COVID-19 prevented them from receiving programming, which included the suspension of dialectical behavioural therapy [79]. Similarly, a study of 16 carceral systems offering opioid agonist treatment (OAT) programs across the USA found that COVID-19 resulted in the downsizing of operations: of the 16 systems surveyed, 10 reduced the scale of their OAT programs, 7 changed their medication dispensation process, which included limiting the frequency of assessments, and 1 discontinued follow-up appointments upon release [80]. A study of blog posts from people in prison in San Vittore, Italy identified a suspension of substance use disorder treatment: "This epidemic suspends the opportunity we had to treat ourselves" [73]. A study of correctional facilities in Ireland for men and women highlighted a lack of treatment for substance withdrawal [81].

> "*Was coming off alcohol in custody. No treatment for alcohol withdrawal Started me on librium in police station. [I] didn't get them in here. [I] didn't receive any support from healthcare—withdrawal off alcohol."*

One study reported a change in health care delivery during the pandemic. A telemedicine program was implemented in North Carolina, USA prisons for specialty care, and 94.0% of patients (453/482) reported a positive overall telemedicine experience [82].

Three studies described a loss of services outside of health care. In an Oregon, USA prison, Getting Out by Going In, a leadership program, Step-Up, a skills training program, and General Education Development (GED) classes were all suspended [79]. The suspension of education programs was also identified in studies in two UK prisons [30, 78].

**Impacts on relationships with family and staff.** Four studies described the impact of the pandemic on incarcerated individuals' relationships with their families. In a men's prison in Scotland, one study participant expressed his concern about how the lockdown impacted his relationship with his family [31].

> *"Personally, am incredibly worried about progression so I can be back with my family supporting them at such hard times."*

Incarcerated men in Scotland noted that suspension of visitation had deteriorated relationships with family. One person stated that the lack of visitation resulted in "a breakdown in [his] family life" [30], and another said that the lockdown was "tearing families apart" [29].

One participant highlighted the use of phone calls as a means to connect with family during the suspension of visitation [79].

> *"They cut off visiting for obvious reasons. It's cut off til further notice, but that's why Telmate [a private company providing inmate phone calls] has been giving us free calls so we can still stay connected with our families because family is a big part of DOC and keeping sane."*

Participants in a study of men's and women's correctional facilities in Ireland similarly cited the value of phone calls and virtual visits with family for mental health [81].

> *Phone calls really helped to take my mind off thing[s], I spoke with my mum and felt much better, I got to sleep that night.*

Even phone access could be problematic, however; one participant in the study in Scotland described his reluctance to use the phone due to concerns about lack of phone sanitation and minimal access to hand sanitizer [31].

Four studies commented on the effects of COVID-19 on relationships between people in custody and staff. One study noted that the pandemic strained relationships between incarcerated individuals and staff, with study participants describing heightened tensions and an increase in conflict [31]. Frustrations associated with the pandemic were expressed as "scuffles with prisoners and staff" and "unnecessary violence" [31]. One participant described the feeling of being a burden to staff, while another felt that staff were more detached from their roles since the start of the COVID-19 pandemic, for example with staff not responding to cell buzzers when individuals required assistance [31]:

> *"Too many officers sit at the desk. No social distancing that equals to what's forced on prisoners. Buzzers are being pressed and on more than one occasion it's taken 30 minutes plus to answer, one was 90 minutes until more people started kicking doors on various landings."*

A study in UK prisons found that relationships between incarcerated individuals and staff had suffered partly because incarcerated individuals blamed correctional staff for COVID-19 restrictions which they perceived as "harsh" and "an additional punishment" [78].

In contrast, three studies found that during the pandemic, staff were increasingly supportive of incarcerated individuals. One participant in Northern Ireland stated that "[isolation] was absolutely terrible, but the staff were lovely. Really helpful was staff" [81]. In a study in Italy, a participant expressed appreciation for the ongoing work of prison workers [73].

> *"Every day our counsellors show up on time and we continue treatment, even if it is reduced. We always see them smiling at us, trying to minimize the problem, giving us hope and bringing us news. They are always present to listen to our problems. . ."*

Similarly, in a study from British Columbia, Canada, peer health mentors supporting people leaving prison reported more compassion from correctional staff towards incarcerated individuals [83].

> *"They're not about punishment right now, and they're really about care and understanding and, you know, really worried about their health."*

**Impacts on mental health.** Several studies explored the impact of the pandemic on mental health. A person in a UK prison described a "worsening situation, feelings of isolation, and the increased weight of time in prison" [31]. People incarcerated in an Italian prison from March to May 2020 described worry, psychological pain, and fear [73]. Isolation emerged as a common theme across studies: "I stay in my cell 24 hours a day, except for those minutes dedicated to phone calls with my loved ones" [73], with study participants describing the impacts of isolation on mental health, exacerbating "feelings of boredom, frustration and stress" as well as suffering [29, 30, 73].

> *"There has been a rise in mental health issues during the lockdown due to the amount of time spent in isolation. There has even been a suicide in [name of prison]."* [67]

Many incarcerated individuals described how isolation exacerbated depression, anxiety, thoughts of self-harm, and suicidal ideation [78, 81].

> *"My depression is coming back. It was really hard not speaking to anyone either on landing or family. I had suicidal thoughts and self-harmed but didn't tell anyone."*

> *"Struggling, have self harm on a number of occasions in secret [. . .] its lonely and depressing especially as I have no one outside prison supporting me anyhow, so now completely isolated so increases my suicide and self-harm thoughts"*

In another study, a participant expressed feeling grateful about being in solitary confinement, as a strategy to prevent transmission of COVID-19 [79].

> *"I definitely think, I mean, this is the one time in my life I've been grateful to be in isolation."*

Another common theme was anxiety surrounding infection risk, with study participants particularly concerned about infection transmission from staff, given that they had limited contact with the outside community otherwise [31].

> *"Let's hope it doesn't get into prison, god help us if it does*!!*"*

In an Oregon, USA prison, study participants had varying levels of concern about infection risk, with 74.2% of participants reporting that they were "either not at all or somewhat worried" that their institution would become infected with COVID-19, while 25.8% were "either pretty or extremely worried" [79]. Many participants indicated that they were not worried about a COVID-19 outbreak in their facility because they had no control over the disease's spread [79].

Three studies identified participants' desire for greater transparency from the prison system regarding the pandemic. In one study, participants described anxiety surrounding a lack of communication from prison officials [31]. Participants reported that they were not provided information about the outside community, while the information they were able to access was outdated [31]. In another study, participants described feeling that newsletters from prison officials were not adequate, as the information within was sparse and not tailored to their specific unit [79]. A study in a Northern Ireland prison found that 71% of respondents felt they had not received the appropriate information, which further added to anxiety [81].

> *"There was an expectation that we should just know things, nothing was communicated to me. Mental [h]ealth has been up and down, feel a panic attack coming on."*

A study of peer health mentors supporting people leaving prison found that COVID-19 had also intensified anxiety around release from custody [83]. Peer health mentors reported that incarcerated individuals were reaching out for increased emotional support [83].

> [They're] grateful that we're still here and we're still answering our phone. And a lot of them just reach out just to talk. 'Cause there's so much unknown out here. And they know every-thing's shutdown and they're, like, 'what's going to happen to me?'

Finally, a study of reported self-harm events in a people in a Swiss prison from 2016 to 2020 found that comparing 2016–2019 to 2020, there was a 57% increase in severe suicide attempts from 4.4/100 people in detention/year to 7.0/100 people in detention/year, and a 57% increase in other self-harm events from 7.9/100 people in detention/year to 12.5/100 people in deten-tion/year [84].

## Discussion

This systematic review identified 62 studies on the health impacts of the COVID-19 pandemic on people who experience imprisonment. Key findings are that incarcerated populations have experienced disproportionately high rates of COVID-19 infection, COVID-19 mortality, and all cause-mortality. The COVID-19 pandemic was associated with worse access to many health and non-health services, increased isolation and feelings of powerlessness, and negative impacts on relationships with family and on mental wellbeing, as well as varied impacts on relationships with staff. There is a lack of data on the health impacts of the pandemic beyond the risk and short-term outcomes of COVID-19 infection itself, and in particular, a lack of data on the experiences of people who experience imprisonment globally and a lack of quanti-tative data for people outside of the USA.

Regarding limitations of individual studies, most studies were at risk of bias in one or more domains examined. Common limitations were the lack of data on the population included and on the setting, the lack of comparator population data, the lack of stratification or adjustment for important risk factors (such as age) for outcomes such as incidence, hospitalization, and death, the lack of denominator data to be able to understand rates in addition to counts, and a lack of clarity regarding methods used to identify outcomes of interest.

There are two main limitations to this review. First, we limited our search to published peer-reviewed studies and published pre-print studies. Since published peer-reviewed and published pre-print studies may have a unique influence on public policy compared with sur-veillance data alone, including these published studies is of value in and of itself, but that not-withstanding, future work could include searches of additional grey literature to enhance the data identified. Second, we updated our search in October 2021, which was over 18 months after the World Health Organization declared SARS-CoV-2 a pandemic on March 13[th], 2020. We appreciate that in the context of a global emergency such as this pandemic, additional scholarly work will emerge in the coming months and years, and that there is often a substan-tial lag from the time of the completion of research to publication. This review serves to sum-marize the literature to date, which is important to inform ongoing pandemic response, recovery planning and implementation, and future emergency preparedness.

The variation in findings across studies for certain important COVID-19 outcomes is strik-ing. For example, percent positivity in outbreaks varied between 2.5% and 90.6%. This varia-tion could reflect true differences, for example differences in prison structures and policies, prevention measures, and population characteristics and behaviours, but could also reflect how the study was conducted, for example over what follow up period data were collected,

who was considered at risk of infection and included in the denominator, and who accessed testing. For studies comparing COVID-19 mortality between people in prison and the general population, the magnitude of the difference in risk varied substantially across studies, and in one study even the direction of association was different. These differences may be due to differences in the populations compared, for example the study that found that people in prison had lower risk of COVID-19 mortality compared with the general population did not control for age [44], or may be due to circumstances at specific points in time in prison settings and in the community, which could be attributable to prevention strategies, epidemiology, or chance.

While this systematic review was designed to be descriptive rather than explanatory, additional work would be valuable to quantitatively summarize aspects of the burden of disease of COVID-19, including evidence for risk factors and risk mechanisms. Further, research is needed to elucidate the impacts of specific COVID-19 response strategies for this population and setting, recognizing the unique opportunity to compare findings across person, place, and time, though this work would notably require considerable data harmonization efforts. Understanding which prevention strategies have reduced the burden of COVID-19 and other harms during the COVID-19 response is necessary to inform effective emergency planning and response, as well as to support ongoing health promotion and health care.

Given the major burden of the COVID-19 pandemic on incarcerated populations, prisons and people who experience imprisonment should be prioritized in ongoing COVID-19 response and recovery efforts [85]. This could involve implementing evidence-based strategies now to mitigate the risks of adverse health impacts of the pandemic, for example interventions to promote mental health and treat mental illness. As part of a broader public health agenda to address the harms of the carceral system [19], future emergency prevention and preparedness efforts should explicitly consider this setting and population. Additional work is needed to understand the impacts of the pandemic on the health status of this population, in both the short and long-term, particularly regarding the experiences of people in prisons including their health care access and quality.

## Supporting information

**S1 Table. Risk of bias for quantitative studies included in a systematic review on the health impacts of the COVID-19 pandemic on people who experience imprisonment.**
(DOCX)

**S2 Table. Risk of bias for qualitative studies included in a systematic review on the health impacts of the COVID-19 pandemic on people who experience imprisonment.**
(DOCX)

**S1 File. Database search strategy.**
(DOCX)

**S1 Checklist. PRISMA checklist.**
(DOC)

## Author Contributions

**Conceptualization:** Hannah Kim, Emily Hughes, Alice Cavanagh, Susan J. Bondy, Fiona G. Kouyoumdjian.

**Data curation:** Hannah Kim, Emily Hughes, Alice Cavanagh, Emily Norris, Angela Gao, Susan J. Bondy, Katherine E. McLeod, Tharsan Kanagalingam, Fiona G. Kouyoumdjian.

**Formal analysis:** Emily Hughes, Alice Cavanagh, Angela Gao, Fiona G. Kouyoumdjian.

**Supervision:** Fiona G. Kouyoumdjian.

**Writing – original draft:** Hannah Kim, Emily Hughes, Angela Gao, Katherine E. McLeod, Fiona G. Kouyoumdjian.

**Writing – review & editing:** Hannah Kim, Emily Hughes, Alice Cavanagh, Emily Norris, Angela Gao, Susan J. Bondy, Katherine E. McLeod, Tharsan Kanagalingam, Fiona G. Kouyoumdjian.

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
