## [Decision Letter · Decision Letter 0]

23 Mar 2022

PONE-D-22-05150The health impacts of the COVID-19 pandemic on people who experience imprisonment: A mixed methods systematic reviewPLOS ONE

Dear Dr. Kouyoumdjian,

Thank you for submitting your manuscript to PLOS ONE. After careful consideration, we feel that it has merit but does not fully meet PLOS ONE’s publication criteria as it currently stands. Therefore, we invite you to submit a revised version of the manuscript that addresses the points raised during the review process.

We look forward to receiving your revised manuscript.

Kind regards,

Seth Blumberg

Academic Editor

PLOS ONE

Journal Requirements:

2. We note that this manuscript is a systematic review or meta-analysis; our author guidelines therefore require that you use PRISMA guidance to help improve reporting quality of this type of study. Please upload copies of the completed PRISMA checklist as Supporting Information with a file name “PRISMA checklist”.

Additional Editor Comments:

Both reviewers and I found the manuscript’s focus on people who experience imprisonment to be compelling. I agree with the reviewers that the mixed-methods approach for conducting the systemic review was thoughtfully planned and well-articulated. Meanwhile, there is also general consensus that the impact of the systemic review gets watered down in the results and discussion as there was not enough interpretation of the findings. The reviewers provide excellent feedback for improving the article in this regard. I look forward to your review in which the reviewer feedback is incorporated.

Also, a few small thoughts as I read the article:

- I found the wording of lines 44-46 of the abstract awkward to interpret.

- Comparison of test positivity is a bit challenging because it depends on who gets test, and when they are test, etc. I think it would help to mention all these nuances so that the wide variability in percentages can be put into place

- Similarly, can you provide contextualization for why covid mortality results may be higher in general population for some studies and prison population for others?

- Vaccine hesitancy vs acceptance evolved as the pandemic progressed. Can that explain any of the variance in vaccination rates?

- The first paragraph of the discussion has several claims that are a bit challenging to abstract from the results. However, with improved presentation of the results, I think the link will be more evident

- The material in the second paragraph of the discussion can probably be incorporated into the results section. (i.e., items #15/#22 of the PRISMA checklist)

Reviewers' comments:

Reviewer's Responses to Questions

**Comments to the Author**

1. Is the manuscript technically sound, and do the data support the conclusions?

Reviewer #1: Yes

Reviewer #2: Yes

2. Has the statistical analysis been performed appropriately and rigorously? 

Reviewer #1: N/A

Reviewer #2: N/A

3. Have the authors made all data underlying the findings in their manuscript fully available?

Reviewer #1: Yes

Reviewer #2: Yes

4. Is the manuscript presented in an intelligible fashion and written in standard English?

Reviewer #1: Yes

Reviewer #2: Yes

5. Review Comments to the Author

Reviewer #1: Thank you for your work! This paper is methodologically sound, well written and framed, the work is thorough, and the studies used were clearly stated. More detailed feedback has been included within the attached review.

Reviewer #2: 1) The authors conducted a well-documented, far-reaching review of articles linked to people with imprisonment in COVID-19. The article provides high-quality, easily accessed information that will be useful to researchers working on COVID-19 in the carceral setting. I appreciated the detailed review of data extraction methods for both the quantitative and qualitative approaches, and the explanation of their move towards a narrative synthesis. I also thought that the Table format detailing the thematic areas as a review was a helpful orientation of the succeeding sections, and that the discussion gave a helpful synthesis of a large amount of information collected. Finally, I certainly agree with the authors that it helps to convey the importance of continued research on health in the carceral setting, and does a good job identifying gaps.

There are several areas where I would like clarification:

- In the introduction, a number of broad statements are made about "prisons" and various response efforts and consequences with relatively few citations. Given that the systematic review encompasses a global population, this section would benefit from further specificity both in describing where the data comes from, and the broader implications. For example: "While many jurisdictions reduced their prison population size through early releases, pardons, diversions, and release on bail or parole [12], there may not have been commensurate increases in discharge planning or community resources to support needs, including for treatment beds and shelter beds. In addition, the transition to remote services and reduced scope and hours of services may have limited access to essential health and social services [13]." How does this extend to the global prison population, given that these citations are each from Canada and from Germany?

- When distinguishing between "published" and "pre-print" literature, are the authors intended to refer to peer-reviewed literature in journals as "published," or all published literature regardless of peer review? Please make this clear in a revision.

- "If studies included data for youth in 131 detention or other subpopulations in addition to adults in prisons" on line 131; who are other subpopulations?

- For the case-fatality, could you be clear why you report some as percentages as some as fractions? Could also put this in methods.

On the whole, I think it would be helpful to clearly characterize the study as a global review of adults experiencing imprisonment. I also think for certain sections, and particularly test positivity, a long section packed with numbers that are difficult to contrast when reading, a graphic would more effectively compare and present very interesting results (and could simultaneously show country, # of people tested, percent positivity, and time point). Related to this, I think further use of comparison and contrast throughout the reporting sections would also strengthen the readability of the article.

Finally, I do have a concern with the exclusion of studies that "described only properties of 143 the virus in prisons, such as reproduction ratio"; the reproduction ratio is the product of contact rate, i.e. human behavior and interventions, susceptibility and other relevant factors, and not inherent to the virus itself. That said, I would agree with the decision to exclude studies focused exclusively on mathematical modelling and not reporting original data.

2) Not applicable, no statistical analysis

3) Data referenced is generated from the published and peer-print literature, and to my understanding all articles are cited in the bibliography and supplementary tables.

4) On the whole, the article is well-written and clearly conveys the ideas. There are a handful of grammatical errors; specific ones identified include lack of capitalization of "Google" on line 122, "in" instead of "on" on line 133, missing a "the" before "most" on line 307.

There are also a couple of where language is unclear: "The high prevalence of health conditions"; which conditions? In the section on test positivity, I would ask that please restrict the use of percentages to percent positive (e.g. "30.5% in a men’s prison in Arkansas in April and May 247 2020, in which 99.2% of 1,661 people in prison were tested") as it becomes very easy to get lost in this section.

6. PLOS authors have the option to publish the peer review history of their article (what does this mean?). If published, this will include your full peer review and any attached files.

Reviewer #1: No

Reviewer #2: No

---

## [Author Response · Author response to Decision Letter 0]

11 Apr 2022

Journal Requirements:

We have reviewed these documents and we think that our manuscript meets all style requirements.

2. We note that this manuscript is a systematic review or meta-analysis; our author guidelines therefore require that you use PRISMA guidance to help improve reporting quality of this type of study. Please upload copies of the completed PRISMA checklist as Supporting Information with a file name “PRISMA checklist”.

We have renamed the file as requested.

Additional Editor Comments:

Both reviewers and I found the manuscript’s focus on people who experience imprisonment to be compelling. I agree with the reviewers that the mixed-methods approach for conducting the systemic review was thoughtfully planned and well-articulated. Meanwhile, there is also general consensus that the impact of the systemic review gets watered down in the results and discussion as there was not enough interpretation of the findings. The reviewers provide excellent feedback for improving the article in this regard. I look forward to your review in which the reviewer feedback is incorporated.

Thank you very much. We have revised the article based on the helpful feedback from reviewers and we think the manuscript is much improved.

Also, a few small thoughts as I read the article:

-I found the wording of lines 44-46 of the abstract awkward to interpret.

We appreciate this comment and we have revised the language.

-Comparison of test positivity is a bit challenging because it depends on who gets test, and when they are test, etc. I think it would help to mention all these nuances so that the wide variability in percentages can be put into place.

We have provided additional detail in the Results and added information regarding this important issue in the fourth paragraph of the Discussion.

-Similarly, can you provide contextualization for why covid mortality results may be higher in general population for some studies and prison population for others?

We have revised relevant content in the Methods section to present data separately when controlling for age and sex, and we discussed this specific finding in the fourth paragraph of the Discussion.

-Vaccine hesitancy vs acceptance evolved as the pandemic progressed. Can that explain any of the variance in vaccination rates?

We think this is unlikely, given that the periods for the three studies that reported vaccination rates were similar: December 2020-February 2021, until March 2021, and until April 2021.

We have added a comment in the Synthesis section of the Methods to note that we reported on study period as well as other factors consistently, to support interpretation of data that may vary over time.

-The first paragraph of the discussion has several claims that are a bit challenging to abstract from the results. However, with improved presentation of the results, I think the link will be more evident.

We think that the revisions have made the findings more clear.

-The material in the second paragraph of the discussion can probably be incorporated into the results section. (i.e., items #15/#22 of the PRISMA checklist)

As we did not conduct a meta-analysis, we do not have information relevant to PRISMA Items 15 and 22, i.e. assessing for publication bias or selective reporting bias (as per https://journals.plos.org/plosmedicine/article?id=10.1371/journal.pmed.1000100).

In the second paragraph of the Discussion, we have summarized the quality of studies at a very high level (i.e. to summarize what was presented in more detail in the Results section including relevant tables), and elaborated on common limitations, which we hope will inform interpretation as well as future work.

Reviewers’ comments:

Reviewer #2: 

1) The authors conducted a well-documented, far-reaching review of articles linked to people with imprisonment in COVID-19. The article provides high-quality, easily accessed information that will be useful to researchers working on COVID-19 in the carceral setting. I appreciated the detailed review of data extraction methods for both the quantitative and qualitative approaches, and the explanation of their move towards a narrative synthesis. I also thought that the Table format detailing the thematic areas as a review was a helpful orientation of the succeeding sections, and that the discussion gave a helpful synthesis of a large amount of information collected. Finally, I certainly agree with the authors that it helps to convey the importance of continued research on health in the carceral setting, and does a good job identifying gaps.

We appreciate these comments. 

There are several areas where I would like clarification:

-In the introduction, a number of broad statements are made about "prisons" and various response efforts and consequences with relatively few citations. Given that the systematic review encompasses a global population, this section would benefit from further specificity both in describing where the data comes from, and the broader implications. For example: "While many jurisdictions reduced their prison population size through early releases, pardons, diversions, and release on bail or parole [12], there may not have been commensurate increases in discharge planning or community resources to support needs, including for treatment beds and shelter beds. In addition, the transition to remote services and reduced scope and hours of services may have limited access to essential health and social services [13]." How does this extend to the global prison population, given that these citations are each from Canada and from Germany?

We have revised the content in the third paragraph of the Introduction to clarify the text and provide additional references.

-When distinguishing between "published" and "pre-print" literature, are the authors intended to refer to peer-reviewed literature in journals as "published," or all published literature regardless of peer review? Please make this clear in a revision.

We have clarified this content in the Search section of the Methods, as well as in the third paragraph of the Discussion.

-"If studies included data for youth in 131 detention or other subpopulations in addition to adults in prisons" on line 131; who are other subpopulations?

We have revised the language to say “populations” rather than “subpopulations.” We specified this criterion since we anticipated (based on prior reviews we have conducted) that we might identify studies that presented data for people in prisons together with people in the community, rather than stratified data for adults in prison.

-For the case-fatality, could you be clear why you report some as percentages as some as fractions? Could also put this in methods.

We have revised the presentation of the data to consistently report case fatality ratios as percentages.

On the whole, I think it would be helpful to clearly characterize the study as a global review of adults experiencing imprisonment. 

Thank you for this suggestion. We have revised content in the title from “people who experience imprisonment” to “adults who experience imprisonment globally.” 

I also think for certain sections, and particularly test positivity, a long section packed with numbers that are difficult to contrast when reading, a graphic would more effectively compare and present very interesting results (and could simultaneously show country, # of people tested, percent positivity, and time point). Related to this, I think further use of comparison and contrast throughout the reporting sections would also strengthen the readability of the article.

We appreciate this suggestion, and we have added figures to summarize content in two sections of the Results (Percent positivity, as you suggested, and Proportion of cases hospitalized) and a table in the Population mortality rates section. We have also added content in the Discussion regarding the comparability of findings across studies.

Finally, I do have a concern with the exclusion of studies that "described only properties of 143 the virus in prisons, such as reproduction ratio"; the reproduction ratio is the product of contact rate, i.e. human behavior and interventions, susceptibility and other relevant factors, and not inherent to the virus itself. That said, I would agree with the decision to exclude studies focused exclusively on mathematical modelling and not reporting original data.

On reflection, we agree that it makes sense to include studies that report on reproduction ratio for people in prison using original data. We have revised the methods section, and reviewed papers for eligibility that reported on reproduction ratio. We ended up including one additional study: Puglisi et al., though we did not include the data on reproduction ratio since it included staff in prisons and our explicit focus is on people who experience imprisonment, not staff.

2) Not applicable, no statistical analysis

This is correct. 

3) Data referenced is generated from the published and peer-print literature, and to my understanding all articles are cited in the bibliography and supplementary tables.

This is correct. 

4) On the whole, the article is well-written and clearly conveys the ideas. There are a handful of grammatical errors; specific ones identified include lack of capitalization of "Google" on line 122, "in" instead of "on" on line 133, missing a "the" before "most" on line 307.

We have revised the text, and we apologize for these errors.

There are also a couple of where language is unclear: "The high prevalence of health conditions"; which conditions? 

We have revised this text in the second paragraph of the Introduction.

In the section on test positivity, I would ask that please restrict the use of percentages to percent positive (e.g. "30.5% in a men’s prison in Arkansas in April and May 247 2020, in which 99.2% of 1,661 people in prison were tested") as it becomes very easy to get lost in this section.

As noted above, we have substantially revised this section, and the data are now presented as percentages consistently, including in Figure 2. 

Reviewer #1:

Summary

Thank you for taking the time to conduct a thoughtful and careful review of the current published/pre-printed literature on COVID-19’s impact on those who have been incarcerated; this is crucial and important work! The methodological approach was transparent and sound, and I applaud the authors for their diligent work. Because the purpose of this article is a review, I believe that there are substantial revisions that are necessary for this to sufficiently serve that purpose to readers. It’s clear that there was a lot of data extraction done, and this is fantastic, but there was minimal synthesis, contextualization, and organization of these findings, which is crucial for providing a thorough and manageable review of literature. While suggested revisions (particularly for the data synthesis and discussion sections) are considerable, much of the content is already in the paper, but could use some reorganizing, reframing, or a deeper synthesis. As you aptly highlighted, there are more studies that will be coming out and more work in this area that will need to be done, and therefore, providing more clear recommendations based on the large breadth of information reviewed (taking into account the sampled studies’ limitations,) would be a wonderful addition to help drive home the value of all this great work. More detailed and specific comments are provided below – thank you for allowing me the opportunity to read your paper, and I hope these comments prove useful to you!

We appreciate these comments and have addressed specific comments as detailed subsequently.

Introduction – Comments

•This section was compelling, concise, well-framed! Below are just a few comments on some areas to further strengthen this section:

o I think some clarification around the risk of COVID-19 being introduced into prisons being high could be useful here. The criminal justice system (including courts and prisons) altered its practices considerably throughout the pandemic, and so acknowledging that even in the presence of substantial changes and policy shifts (e.g., prisons massively changing how/when people moved, implementing lockdowns, etc.), it was still impossible to effectively halt the introduction of COVID-19 into prisons.

We appreciate this comment, but we think this type of statement would be out of scope for this review and isn’t directly supported by what we found.

o Clarifying point about addressing increases in drug supply is advised, (e.g., that this is specific to individuals upon release.) If you want to address use in prison more explicitly, suggesting the potential for an increase in relapses (or new use) would be warranted and follows from the statement about worsened mental health for those who have been incarcerated during the pandemic.

We have added content to this section in the third paragraph of the Introduction to clarify that changes in the illicit drug supply may impact people both while in prison and in the community post-release.

Eligibility Criteria – Comments

•This section was thorough and clear, and your methodological approach was sound! Your clarity and transparency in how use language is especially appreciated! I want to especially thank you for bringing up the point about policies not necessarily aligning with how prisons are run in practice! Below are just a few comments on some areas to further strengthen this section:

o Just highlighting a really small typo (nothing major): “The first criterion was a focus adults…”

We have fixed this mistake.

Characteristics of Included Studies – Comments

•Table 1 is incredibly helpful and a great addition to this paper! Below are just a few comments on some areas to further strengthen this section:

o While this is only a very minor point, if the table was created in such a way that allows you to easily resort it, that might be very helpful for readers to understand the distribution of studies in your overall sample more easily. I think either resorting based on either time or study location could be particularly helpful. 

We have revised the formatting of the table so the studies are now presented by country.

Risk of bias – Comments

•Thank you for your transparency in addressing the quality of the studies included! Elaborating on the domains where quality indicators were not met would be helpful (e.g., do they all tend to be different, or are there patterns amongst which indicators are not met?) Knowing this information would be helpful for informing the interpretation of results.

We have discussed common limitations in the second paragraph of the Discussion.

Data Synthesis – Burden of COVID infection – Comments

•This section is very thorough! However, I found myself getting very bogged down by a lot of numbers and felt that synthesis, organization, and interpretation of findings was lacking. I also found myself having trouble with interpretation of these figures given the information about study quality (i.e., might the bias be impacting reported figures, and if so, do we expect these numbers to be under/over-estimates? Or maybe not representative?) Below are some specific comments on potential routes to help strengthen this section:

o There are a lot of different numbers for incidence rates/percent positivity presented in this section, which is excellent, but I found myself getting a bit bogged down in the details. Graphs here I think could be particularly helpful and given the fact that the number of figures/tables are not limited, this is strongly advised for helping readers be able to make sense of all of these figures. Then, the written sections can focus more heavily on synthesis. For example, instead of listing something like “Study A found Incident Rate A in Time A and Location A,” starting with something like, “The range in cumulative incidence rates ranged from x to y, with the lowest being in (Time B/Location C, etc.)” 

We appreciate this suggestion. As noted above, we have shifted data from the text into two figures and a table, and in several places we have added content to summarize ranges. We think that these revisions have substantially improved the paper.

o For the figures, even if it’s just a bar graph, ordered by date (when available) with location-specific details included (and patterns/colors associated with location potentially,) with both prison-specific and overall population-specific side-by-side bars. Then the text can focus on describing how much higher these figures were in-prison, compared to respective populations, over time/space. That or a table could be helpful!

As above, we have made changes as suggested. 

o As mentioned above, quality of studies needs to be explicitly woven into interpretation so that we can more accurately and appropriately interpret the figures reported from these studies. I do not think this necessarily means calling out the studies where quality indicators were not met.

We have included content on study size and specified study period, location, and facility type (when applicable) to support interpretation of the internal and external validity of the studies. 

Discussion – Comments

•I appreciated the transparency about how early this study is, as this was important to acknowledge, and is helpful! In this section, I was really hoping for a lot more in terms of synthesis, recommendations, and limitations. Below are a few more specific suggestions that may help support this section: 

o Limitations: I was hoping to see more discussion of what seemed to be some geographic-specific differences in the types of studies conducted. For example, the U.S. seemed to focus a lot more on the epidemiological data (e.g., incidence rates, percent positivity, etc.), while data outside of the U.S. – namely Europe – seemed to be the primary source of qualitative data. This is crucial, and likely reflects considerable differences in penological culture.

We have added a comment in the Characteristics of included studies section that only 1 qualitative study was conducted in the USA. We also added a comment in the first paragraph of the Discussion to specify that there is a lack of data on the experiences of people globally and a lack of quantitative data for people outside of the USA.

o I would love to see more on recommendations! You (rightly!) suggest individuals who are incarcerated need to be prioritized in handling COVID-19 and other public health emergencies. How do we accomplish this? What future studies need to be conducted, based off of what you found? Did it seem that some places were doing better than others in terms of reducing the burden of COVID-19 on its incarcerated populations? Even just highlighting this is important – more work can be done in future studies that could illuminate why this might have been the case. Especially in regards to mental health of residents in prisons – what can be done to support these individuals while still keeping their risk of infection as low as possible (e.g., increasing access to tablets/phone calls, etc.)?

We have added further content in the Discussion, e.g. regarding the need for research to quantitatively summarize the COVID-19 burden of disease and the effectiveness of specific interventions. Based on the nature of the synthesis we conducted as well as the lack of specific data and the lack of comparable data on many outcomes, we don’t think we have sufficient information to be able to comment on whether some places did better than others. We have acknowledged in the final paragraph of the Discussion the importance of implementing interventions now that could mitigate the adverse health impacts of the COVID-19 pandemic in this population, and specifically noted the need to implement strategies to promote mental health and treat mental illness.

---

## [Decision Letter · Decision Letter 1]

1 May 2022

PONE-D-22-05150R1The health impacts of the COVID-19 pandemic on adults who experience imprisonment globally: A mixed methods systematic reviewPLOS ONE

Dear Dr. Kouyoumdjian,

Thank you for the thoughtful set of revisions, which the reviewers expect will improve the readability and impact of this important work. I do not foresee any barriers to eventual acceptance, but for now I am labeling it as a minor revision so that reviewer #1's minor comments can be considered.

We look forward to receiving your revised manuscript.

Kind regards,

Seth Blumberg

Academic Editor

PLOS ONE

Journal Requirements:

Reviewers' comments:

Reviewer's Responses to Questions

**Comments to the Author**

1. If the authors have adequately addressed your comments raised in a previous round of review and you feel that this manuscript is now acceptable for publication, you may indicate that here to bypass the “Comments to the Author” section, enter your conflict of interest statement in the “Confidential to Editor” section, and submit your "Accept" recommendation.

Reviewer #1: All comments have been addressed

Reviewer #2: All comments have been addressed

2. Is the manuscript technically sound, and do the data support the conclusions?

Reviewer #1: Yes

Reviewer #2: Yes

3. Has the statistical analysis been performed appropriately and rigorously? 

Reviewer #1: N/A

Reviewer #2: N/A

4. Have the authors made all data underlying the findings in their manuscript fully available?

Reviewer #1: (No Response)

Reviewer #2: Yes

5. Is the manuscript presented in an intelligible fashion and written in standard English?

Reviewer #1: Yes

Reviewer #2: Yes

6. Review Comments to the Author

Reviewer #1: Thank you for revising your manuscript for review! This is important work, and I believe this paper will be helpful for highlighting how imperative it is to better understand the impact of public health emergencies in the carceral context. 

I believe the authors have done an excellent job on the revision, with tables and figures that helpfully summarize the information and make the manuscript's content more accessible to readers. Limitations and interpretation were bolstered to help address future outlets for research and the limits to what was available at the time of this synthesis. 

I think the Discussion section may still benefit from small edits, namely a stronger stance on the necessity of future research. The authors appropriately identify that there are a variety of reasons that are likely (individually, but also collectively,) contributing to the wide variation in COVID-19 positivity rates and other factors related to the spread of the virus. As the authors identified, published studies may have a unique influence on public policy, and therefore, given their manuscript will hopefully be published (as I will recommend here,) developing this section and more concretely suggesting future avenues could be particularly useful.

Specifically, identifying how to tease out the differences and contributing factors is crucial to understanding the toll of the virus and how to alleviate the burden of health effects on incarcerated populations. The Discussion talks about potential follow-ups as valuable, though they are necessary given the appropriately strong final recommendation that incarcerated individuals need to be thoughtfully considered and properly supported, particularly during public health emergencies. 

In all, I am thankful to the authors for the manuscript and the valuable summary of literature and will recommend its publication. Thank you!

Reviewer #2: Thank you for your thoughtful responses and addressing of comments. I believe it is a very strong manuscript with critical and timely information for researchers, policymakers, and people living and working in prisons, and have no further feedback.

7. PLOS authors have the option to publish the peer review history of their article (what does this mean?). If published, this will include your full peer review and any attached files.

Reviewer #1: No

Reviewer #2: No

---

## [Author Response · Author response to Decision Letter 1]

4 May 2022

See attached Response to Reviewers.

---

## [Editor Report · Decision Letter 2]

10 May 2022

The health impacts of the COVID-19 pandemic on adults who experience imprisonment globally: A mixed methods systematic review

PONE-D-22-05150R2

Dear Dr. Kouyoumdjian,

We’re pleased to inform you that your manuscript has been judged scientifically suitable for publication and will be formally accepted for publication once it meets all outstanding technical requirements.

Kind regards,

Seth Blumberg

Academic Editor

PLOS ONE

Additional Editor Comments (optional):

Thank you for your thoughtful incorporation of reviewer feedback. Congratulations on a job well done!

Reviewers' comments:

N/A

---

## [Editor Report · Acceptance letter]

12 May 2022

PONE-D-22-05150R2 

The health impacts of the COVID-19 pandemic on adults who experience imprisonment globally: A mixed methods systematic review 

Dear Dr. Kouyoumdjian:

I'm pleased to inform you that your manuscript has been deemed suitable for publication in PLOS ONE. Congratulations! Your manuscript is now with our production department. 

Kind regards, 

on behalf of

Dr. Seth Blumberg 

Academic Editor

PLOS ONE